# Frontotemporal Dementia, Where Do We Stand? A Narrative Review

**DOI:** 10.3390/ijms241411732

**Published:** 2023-07-21

**Authors:** Annibale Antonioni, Emanuela Maria Raho, Piervito Lopriore, Antonia Pia Pace, Raffaela Rita Latino, Martina Assogna, Michelangelo Mancuso, Daniela Gragnaniello, Enrico Granieri, Maura Pugliatti, Francesco Di Lorenzo, Giacomo Koch

**Affiliations:** 1Unit of Clinical Neurology, Neurosciences and Rehabilitation Department, University of Ferrara, 44121 Ferrara, Italy; emanuelamaria.raho@edu.unife.it (E.M.R.);; 2Doctoral Program in Translational Neurosciences and Neurotechnologies, University of Ferrara, 44121 Ferrara, Italy; 3Neurological Institute, Department of Clinical and Experimental Medicine, University of Pisa, 56126 Pisa, Italy; 4Institute of Radiology, Department of Medicine, University of Udine, University Hospital S. Maria della Misericordia, Azienda Sanitaria-Universitaria Friuli Centrale, 33100 Udine, Italy; 5Complex Structure of Neurology, Emergency Department, Fondazione Istituto di Ricovero e Cura a Carattere Scientifico Casa Sollievo della Sofferenza, 71013 San Giovanni Rotondo, Italy; 6Centro Demenze, Policlinico Tor Vergata, University of Rome ‘Tor Vergata’, 00133 Rome, Italy; 7Non Invasive Brain Stimulation Unit, Istituto di Ricovero e Cura a Carattere Scientifico Santa Lucia, 00179 Rome, Italy; 8Nuerology Unit, Neurosciences and Rehabilitation Department, Ferrara University Hospital, 44124 Ferrara, Italy; 9Iit@Unife Center for Translational Neurophysiology, Istituto Italiano di Tecnologia, 44121 Ferrara, Italy; 10Section of Human Physiology, Neurosciences and Rehabilitation Department, University of Ferrara, 44121 Ferrara, Italy

**Keywords:** frontotemporal dementia (FTD), primary progressive aphasias, behavioural variant, neurodegenerative dementias, biomarkers, non-Alzheimer’s disease dementias, tau, C9orf72

## Abstract

Frontotemporal dementia (FTD) is a neurodegenerative disease of growing interest, since it accounts for up to 10% of middle-age-onset dementias and entails a social, economic, and emotional burden for the patients and caregivers. It is characterised by a (at least initially) selective degeneration of the frontal and/or temporal lobe, generally leading to behavioural alterations, speech disorders, and psychiatric symptoms. Despite the recent advances, given its extreme heterogeneity, an overview that can bring together all the data currently available is still lacking. Here, we aim to provide a state of the art on the pathogenesis of this disease, starting with established findings and integrating them with more recent ones. In particular, advances in the genetics field will be examined, assessing them in relation to both the clinical manifestations and histopathological findings, as well as considering the link with other diseases, such as amyotrophic lateral sclerosis (ALS). Furthermore, the current diagnostic criteria will be explored, including neuroimaging methods, nuclear medicine investigations, and biomarkers on biological fluids. Of note, the promising information provided by neurophysiological investigations, i.e., electroencephalography and non-invasive brain stimulation techniques, concerning the alterations in brain networks and neurotransmitter systems will be reviewed. Finally, current and experimental therapies will be considered.

## 1. Introduction

Frontotemporal lobar degeneration (FTLD) refers to the set of neuropathologic changes that occur at the cerebral cortex of the frontal and temporal lobes, clinically responsible for major neurologic syndromes that may have some degree of overlap between them. These may include frontotemporal dementia (FTD), a progressive neurodegenerative disorder that manifests mainly with alterations in behaviour, language, or executive functions in general. The term FTD encompasses several clinical manifestations, which include the behavioural variant (bvFTD), the semantic and non-fluent variants of primary progressive aphasia (svPPA and nfvPPA), the right lobe variant (rtvFTD), and FTD associated with motorneuron disease (FTD-MND). FTD-related disorders may also include two tau-deposition-associated neurodegenerative diseases, cortico-basal syndrome (CBS) and progressive supranuclear palsy (PSP), which can show, in their clinical course, signs of frontal lobe dysfunction [1]. The aim of this review is to highlight the salient characteristics of this still underdiagnosed pathology based on recent evidence in the literature. In the field of FTD, many advances have been made since, in 1892, the Czech neurologist Arnold Pick described the first patient with progressive speech impairment and left temporal lobe atrophy. Since that time and the subsequent identification of intraneuronal inclusions known as “Pick’s bodies”, it was to take almost a century for the first diagnostic criteria for FTD language variants to be postulated by Mesulam et al. [2] and for the first three classic clinical manifestations of the condition to be identified [3]. The diagnostic criteria in use today were further revised in the 2000s, as shown below.

## 2. Epidemiology

Until about a decade ago, the diagnosis of FTD was significantly underestimated, until efforts were made to achieve a better characterisation of the clinical phenotypes of the disease and in the identification of biomarkers that would exclude the diagnosis of Alzheimer’s disease (AD). This process has currently increased the incidence of FTD, as pointed out in a recent retrospective cohort study conducted by Logroscino et al. [4]. This study, the Frontotemporal Dementia Incidence European Research Study (FRONTIERS), aimed at assessing the incidence of FTLD across European countries, found an estimated annual incidence rate for FTLD in Europe of 2.36 cases per 100,000 persons-year, which, although still rare, is higher than previously recognised. Nowadays, FTD remains a cause of early onset dementia, which is expected to increase in prevalence as the population ages. Among pure neurodegenerative forms, it is the most common type of cognitive decline in people under the age of 65, and among all age groups, it represents the third most common form of dementia after AD and Lewy body dementia (LBD) [5]. FTD is typically diagnosed around the fifth decade, with a mean age of onset of 56 years. However, cases of FTD in the second decade have been reported, with about 13% of individuals developing it before the age of 50. Frequently, in the youngest, FTD may be misdiagnosed with some psychiatric disorders, such as schizophrenia, bipolar affective disorder, and major depression, especially at the onset [6]. A predominance in the male population has been reported for the bvFTD and svPPA. In contrast, the nfvPPA seems to prevail in females. The bvFTD appears to be the most common type, comprising about 70% of all FTD cases [7]. Regarding disease progression and survival, FTD patients would appear to have reduced life expectancy and more rapid progression of cognitive decline compared with those affected by AD. Life expectancy also appears to be closely correlated with FTD subtype and can range from about 3 years for patients with FTD-MND to more than 12 years for patients with svPPA [8]. Population and clinical studies have shown that FTD is familial in 30–50% of cases, predominantly with an autosomal dominant mode of transmission. Therefore, a number of responsible gene mutations have been identified, such as the following: chromosome 9 open reading frame 72 (*C9orf72*), in which psychosis at presentation is not uncommon and which is mostly associated with amyotrophic lateral sclerosis (ALS), bvFTD, and FTD-MND; microtubule-associated protein tau (*MAPT*), associated with familiar parkinsonism with FTD; valosin-containing protein 1 (*VCP1*), a familial disorder characterised by inclusion body myopathy with osteolytic bone disease and FTD; chromatin-modifying protein 2B (*CHMP2B*); and progranulin (*GRN*), strongly associated with bvFTD and nfvPPA. *MAPT*, *GNR*, and *C9orf72* are the most common genes responsible for autosomic dominant inheritance of FTD. Among the syndromes belonging to the FTD spectrum, svPPA is the only one that least commonly exhibits a genetic mode of transmission [8,9]. Among nongenetic risk factors for developing FTD, several hypotheses have been made. It is currently believed that an important role may be played by neuroinflammation, and in this regard, studies have been carried out relating blood levels of tumour necrosis factor (TNF) and neurodegeneration associated with FTD. It would also appear that subjects with autoimmune disorders, including thyroid disease, are more vulnerable to the development of FTD, non-thyroid autoimmune disorders in particular being twice as common in subjects with svPPA than in the general population [6,7]. Other hypotheses regarding environmental influence would seem to include head trauma, a previous diagnosis of specific learning disorders regarding the variants with language impairments, and right hemispheric dominance, due to a predominant involvement of left-handed individuals in the development of svPPA [5].

## 3. Clinical Manifestations

### 3.1. bvFTD

BvFTD, characterised by the gradual onset and progression of behavioural alterations, represents the most frequent clinical manifestation of the disease, with an average age of onset around the end of the fifth decade [4]. The clinical picture typically may show the presence of six different pivotal elements, at least three of which are necessary for the clinical diagnosis of bvFTD, as will be shown below: disinhibition, apathy, loss of empathy, altered eating behaviour, stereotyped attitude, and impaired executive functions [10]. Disinhibition is generally evident in the enactment of inappropriate behaviours, such as use of offensive terms during conversations, sexually explicit attitudes, and impulsivity. Disinhibited patients may show a tendency for ease in releasing personal information to strangers, a childish attitude, use of offensive and inappropriate language, heterodirected aggression, violation of interpersonal barriers, and more generally a loss of inhibitory control and social decorum. It is not uncommon to observe in these individuals a full-blown impulse dyscontrol syndrome with enactment of illegal behaviours, substances abuse, including excessive consumption of alcohol and tobacco, gambling, and overspending [11,12]. Apathy, on the other hand, is often an early manifestation of the condition and is characterised by a loss of interest in social and non-social activities, indifference, reduced drive in movements, loss of interest in goal-directed activities, and a tendency towards social withdrawal. Apathy can be distinguished into three subtypes: cognitive, autoactivation/behavioural, and affective-emotional. Cognitive apathy manifests itself with a reduction in planning activities and voluntary actions; autoactivation/behavioural apathy presents with a reduction in self-activating thoughts and behaviours; and finally, affective-emotional apathy is represented by a loss of interest in activities that were previously considered pleasurable and stimulating [13]. This clinical manifestation, at first glance, may be difficult to distinguish from a major depression; however, in the case of apathy, there is a sudden onset of loss of interest in activities of daily life, regardless of the individual’s mood status. Apathy, in individuals with bvFTD, is also generally not associated with suicidal behaviour or ideation and patients more frequently show reduced motor activity [1,13]. Furthermore, bvFTD patients may exhibit perseverative, stereotyped, compulsive, and ritualistic behaviours. Stereotypic behaviours, which can involve both spoken language and motor aspects, on the motor side may be simple, such as tapping, picking, lip smacking, or rubbing, or they may involve more complex sets of actions. An extreme, rarely observed in this context, is hyperreligiosity [14]. Other pivotal elements of bvFTD are hyperorality, manifested by a tendency to orally explore objects placed in the surrounding environment, and changes in eating habits, with increased food intake and a preference for sweets. In more advanced stages, a tendency to ingest nonedible objects can also be observed. The alterations in eating behaviour, which often result in a binge-eating pattern, lead to exposure to all the risk factors associated with weight gain, such as alterations in glucose metabolism, lipid balance, and increased vascular risk, demonstrating how FTD represents a disorder of complex management not only in the neurological setting [15]. Additional symptoms of this disorder may present as frank executive dysfunction, with easy distractibility, alterations in sustained attention span and working memory, reduced mental flexibility, planning, and problem-solving skills, and verbal fluency. In the forms of bvFTD overlapping motorneuron disease (FTD-MND) and PSP or CBS, typical motor dysfunctions associated with these syndromes may be concomitant. A peculiar feature of patients with bvFTD who have *C9orf72*-expanded repeat mutations is the frequent occurrence in the early stages of the disease of psychotic features, multimodal hallucinations, bizarre or somatic delusions, and the rare Diogenes syndrome, a condition manifested by the loss of interest in the physical, hygienic, and mental care of oneself [16]. Finally, recent evidence has sometimes demonstrated the association of FTD with autonomic dysfunction, particularly altered pain perception and thermoregulation, and sleep alterations, which suggest a possible hypothalamic involvement, as in the case of altered eating habits [15]. A core element that deserves separate mention in the context of the bvFTD is the impairment of social cognition. The latter is understood as the set of complex skills necessary for involvement in social interactions or, more generically, the set of cognitive abilities needed to interpret oneself in relation to others and the environment. These include, for example, empathy, the ability to read others’ emotions, mentalising about the state of mind of others (theory of mind, ToM), and using these skills to achieve an appropriate behaviour in social contexts. More generally, social cognition also includes the skills of recognising emotions through visual and auditory stimuli, understanding sarcasm, awareness of the possible negative consequences of engaging in socially inappropriate behaviour, and knowledge of moral and social norms [17,18]. Several brain structures have been shown to be responsible for the regulation of these functions; in particular, considerable attention has been paid to the role played by the orbitomedial frontal cortex, anterior insula, and amygdala, particularly in the context of the right hemisphere. Particular attention has also recently been paid to a specific population of von Economo neurons, mainly represented at the level of the frontoinsular cortex and right cingulate gyrus, which would appear to be specialised in the regulation of social cognition and which would be early and diffusely impaired in patients with bvFTD [19]. Specifically, clinically, bvFTD patients present with reduced empathy, inability to solve social dilemmas, and difficulty in making moral judgments and decisions in social contexts. Special emphasis has been placed on the difficulty of these individuals in recognising emotions, a phenomenon on which many studies have recently focused. The patient affected by bvFTD would, therefore, have difficulty associating facial expressions with related emotions, and this phenomenon would also appear to be impaired in relation to acoustic stimuli, with inability, for example, to distinguish laughter from crying. It has been observed that the recognition of negative emotions, in particular, mainly appears to be impaired, with a significant impact on the quality of life of caregivers as well [20,21].

#### bvFTD Neurological Examination and Neuropsychological Profile

The neurological examination is generally normal, except for the possible presence of frontal release reflexes (e.g., glabellar reflex, grasping, and palmo-mental reflex). However, during the clinical evaluation, an inappropriate, overly impulsive, or conversely, apathetic attitude, enactment of infantile behaviours, evidence of poor personal hygiene, absence of insight, and environmental dependence (i.e., the tendency to constantly manipulate objects placed in the surrounding environment) may become apparent [22]. Special attention should be paid to looking for any signs of motor neuron damage, such as fasciculations, muscle atrophy, spasticity, hyperreflexia, or elements indicative of possible CBS (i.e., asymmetric bradykinesia-rigidity-dystonia), or even PSP (slowness of saccades, limitations in upwards gaze, rigidity, and unsteadiness in walking), in order to rule out possible overlap syndromes [23]. An essential role in distinguishing FTD from other forms of dementia is played by neuropsychological assessment, which must be aimed at investigating multiple cognitive domains and be anatomically oriented towards specific brain structures. Moreover, it is fundamental to evaluate not only the mere quantitative score aspects obtained on neuropsychological tests but also and above all the qualitative aspects of performance, which also consider the attitudes enacted by the patient and the type of errors made. In particular, during the course of the test, patients may appear disinterested, poorly motivated, easily distracted, and impulsive, with difficulty in waiting until the examiner has finished explaining how to perform the test [24]. The cognitive profile of these patients in the early stages may be almost within the normal range, particularly in cases of atrophy mainly involving the right temporal lobe [25]. Language is generally spared, as are visuospatial and praxic functions and memory, although there may sometimes be selective impairment of episodic memory [26]. Rare cases of altered speech patterns with stereotypies, echolalia, and loss of spontaneity, up to mutism in the more advanced stages, have also been reported [27]. What is found to be early and mainly impaired are executive functions, understood as planning skills, organisation, mental flexibility, and complex thinking in general [28]. Neuropsychological tests that can adequately identify such dysfunctions are represented, for example, by those that can assess set-shifting abilities, (e.g., the trail-making test (TMT), the digit span backwards test, or those of verbal and nonverbal fluency), inhibition (such as the Stroop test), and the frontal assessment battery (FAB) as a whole, as well as tasks that assess error sensitivity and decision-making ability [17]. One particular aspect that has emerged during the assessment of these patients when performing verbal, phonemic, or semantic fluency tests is the increased occurrence of errors given by a higher number of repetitions compared to other patients [29]. Figure copying may also be impaired because of poor organisation skills in task execution. An emerging chapter is the development of tests that highlight the existence and extent of impairment in the area of social cognition in these patients. Some of them involve, for example, facial expressions recognition, which involves pairing a facial expression with a label that reports a specific emotion [30]. All of these tests highlight a general impairment in emotion recognition ability, especially affected in relation to negative emotions [31]. In particular, the Social Cognition and Emotional Assessment (SEA) and the mini-SEA have been shown to adequately discriminate between bvFTD and other conditions, such as AD and major depressive disorder [32]. There are also tasks that specifically aim to highlight any deficits in ToM, such as some that use visual cartoons in which patients have to understand the mental state of others [33,34].

### 3.2. Language Variants of FTD (PPAs)

PPAs comprise three clinical variants, of which only two, svPPA and nfvPPA, fall within the spectrum of FTD. In contrast, the logopenic variant of PPA (lvPPA) is more commonly associated with AD.

#### 3.2.1. SvPPA

SvPPA is characterised by a mean age of onset around 60 years and slower progression than bvFTD [35]. In this disorder, cortical atrophy at onset is typically asymmetrical, primarily affecting the left or right anterior temporal lobe, and with only subsequent and progressive extension to contralateral homologous regions and the ipsilateral orbitofrontal cortex. This progression occurs slowly over about three years, and most patients have left anterior temporal lobe involvement in the early stages (around 70% of patients) [36]. In the latter case, the pivotal clinical element is represented by an alteration in language, which appears fluent but meagre in content, full of semantic paraphasias, circumlocutions, and anomie. The patient therefore manifests a loss of the ability to recognise the meaning of words, at first in a sectorial manner for less commonly used ones, and then progressively for more frequently used terms as well. In contrast, verbal fluency and syntactic and grammatical aspects of language are typically spared [37]. In the case of the main involvement of the right anterior temporal lobe, on the other hand, the appearance of behavioural alterations can be observed, similarly to bvFTD, with disinhibition, alterations in eating habits, appearance of obsessive–compulsive behaviours, and poor insight. Patients generally show emotional detachment, easy irritability, tendency to social isolation, and alterations in sleep, appetite, and libido [38]. Also in this case, as already observed for bvFTD, a ToM deficit and a difficulty in reading others’ emotions can be found, caused by dysfunction of the right amygdale [17]. In the case of both left and right lobe involvement, it is also possible to observe the development of new distinct interests, particularly in artistic skills, such as drawing, painting, or music [39].

#### 3.2.2. svPPA Neuropsychological Profile

On the neuropsychological side, the main aspects of the condition can be well evidenced by the use of tests that aim to assess different aspects of language. Specifically, the obvious pivotal element will be the frequent presence of anomie, a classic deficit in the comprehension of single words, and the frequent use of “filler” words, in the context of an overall spared verbal fluency. More difficulty is typically evident in semantic fluency tests than in letter fluency tests, and sentence comprehension appears to be better than single word comprehension [2,40,41]. Another aspect that is evident in patients with svPPA is reading and writing difficulties related to the comprehension disorder, termed surface dyslexia and surface dysgraphia, respectively, with a tendency for the commission of regularisation errors when reading aloud or writing under dictation. In these cases, in fact, irregular words are read or spelled according to letter-sound rules [39,42]. In practice, the most useful tests in these cases are those of confrontation naming, single word comprehension, category fluency, and word-picture matching, including, for example, the palm and pyramids task and the Boston naming test [1,43]. Finally, in the case of rare mesial temporal lobe involvement, an episodic memory disorder can be observed that appears in some ways opposite to that of AD, with better recall for recent events and loss of more remote autobiographical memories [22]. Visuospatial and executive functions are generally found to be preserved [37].

#### 3.2.3. nfvPPA

nfvPPA represents the second most frequent clinical manifestation of FTD, affecting about 25% of patients. The average age of onset is around 60 years, and it is the most heterogeneous of the disorders that fall within the FTD spectrum [37,44]. In nfvPPA, the central element is impaired speech, which becomes progressively nonfluent, laboured, hesitant, characterised by reduced output and sentence length, and increased phonemic errors. The recurrence of grammatical errors in spontaneous language, such as omissions of prepositions, articles, conjunctions, errors in the choice of verb tenses, and errors in subject/verbs agreement, is common [41,45,46,47]. The clinical elements typically evident in patients with nfvPPA are, therefore, represented by agrammatism and apraxia of speech, which is a difficulty in the speech motor programming associated with a deficit in verbal articulation. The presence of these elements, in association or not with each other, allows patients to be distinguished into two groups, one comprising those presenting with apraxia of speech and agrammatism simultaneously, and the other, in which subjects present only with agrammatism [48]. Thus, in the early stages of the disorder, the hallmarks are impairment in speech, writing, and reading, while verbal comprehension is typically spared, except for sentences with more complex grammatical structure [45]. Other language alterations that may become apparent include a distortion of prosody, a reversal, typically in the early stages, of binary terms (e.g., “yes” instead of “no” or “him” instead of “her”), and a tendency to answer questions with “stock” phrases (e.g., “I don’t know”), in order to avoid spontaneous language as much as possible [41,46]. In the more advanced stages, it is common to observe the appearance of mutism and a more pronounced difficulty in verbal comprehension and execution of complex orders [49]. Other manifestations beyond alterations in language may be represented by frank motor deficits in overlap syndromes with PSP or CBS or by the appearance of more subtle motor disturbances, such as a hindrance in fine movements, when the pathology is more extensive in the dominant frontal lobe [50,51]. On the behavioural side, sometimes in advanced stages, behavioural alterations can be found, such as apathy or disinhibition, although less frequently than in bvFTD [52,53]. As in the latter, moreover, some patients, as the disease progresses, may begin to cultivate new interests in visual or verbal arts and music as a result of asymmetric degeneration of the anterior temporal lobes. This phenomenon is probably explained by a gradual loss of function in the dominant temporal lobe, which would lead to a remodelling and increased expression of the functions of the posterior structures of the nondominant hemisphere [54]. Concerning social cognition, a smaller deficit in socio-emotional functioning can be observed compared to svPPA, with an apparent selective deficit in the interpretation of emotions from vocal prosody [17].

#### 3.2.4. nfvPPA Neuropsychological Profile

Neuropsychological evaluation appears to be complex and should first include assessment of the possible presence of apraxia of speech and its severity, as the latter may limit the remaining examination. It is made more complex also by the presence of reduced spontaneous verbal output and agrammatism, assimilated with psychomotor slowing, if present [37]. The neuropsychological profile generally shows the presence of nonfluent language, grammatical errors, with difficulty in word finding, use of circumlocutions, and relative preservation of comprehension [1,37]. Of note, naming and word-finding deficits are common to all forms of PPA, but in nfvPPA, patients do not have difficulty with repetition of sentences, understanding their meaning, semantic associations, or writing or reading irregular words [43,46,47]. The frank presence of anomie is also rare [41]. Of note, no test appears to be adequately sensitive or specific to highlight the patient’s grammatical skills. In this regard, general conversational cues, such as asking the patient about his or her hobbies or occupation, may be useful, as well as assaying parts of written language, such as emails, which often highlight errors that include word order or functional morphemes. Finally, in some patients, verbal descriptions of a figure may be useful [37]. In addition to language disorders, mild deficits in executive functions, particularly verbal fluency, working memory, and set shifting, may coexist, while episodic memory and visuospatial functions are generally spared, although poor planning during visuospatial tasks may sometimes be evident [55].

#### 3.2.5. lvPPA

In contrast, the lvPPA, as described earlier, does not fall within the spectrum of FTD, being in most cases related to AD neuropathology (i.e., β-amyloid and p-tau) and linked to focal initial degeneration at the left temporo-parietal junction. Clinically, patients with lvPPA present with minimal verbal output and intermittent disruption in fluency, with the sparing of grammatical and motor aspects of language. Phonological errors are common, as are impaired single-word retrieval in spontaneous speech and confrontational naming and impaired repetition [56,57,58]. The most widely credited hypothesis underlying the clinical implications of lvPPA would appear to be a deficit in auditory–phonological short-term memory, which would contribute to most of the verbal disturbances, since immediate verbal recall is a key factor in the repetition of sentences and comprehension of longer, more complex sentences [39,59]. The lvPPA differs from svPPA essentially in preserving the ability to comprehend single words and object recognition. It appears similar to nfvPPA with regard to the slowdown in spontaneous speech and the presence of hesitations, but the two forms of PPA differ from each other on the basis of qualitative elements; indeed, pauses for word finding appear in lvPPA, with simple but syntactically correct expressions and absence of motor speech deficits, unlike in nfvPPA [37].

### 3.3. rtvFTD

The rtvFTD appears to be a neurodegenerative disorder with a cognitive, behavioural, and language impairment profile quite distinct from the other FTD variants and AD, with its own unique pattern of temporal lobe atrophy [60,61]. The concept of rtvFTD is relatively recent and in some ways still appears equivocal. Indeed, the diagnosis of rtvFTD is still difficult, making the condition not easy to distinguish in its early stages from bvFTD and AD [62]. Moreover, based on the atrophy pattern on neuroimaging, the syndrome has long been regarded as a variant of svPPA. However, to date, it is considered a distinct entity with its own peculiar profile [63]. Patients in whom cortical atrophy predominantly affects the right temporal lobe clinically present with behavioural alterations, such as disinhibition, obsessive personality, prosopagnosia, spatial disorientation, and episodic memory disturbances [64,65,66]. The most frequent clinical manifestations at onset include personality alterations and difficulty in recognising familiar faces, the latter because of a predominant pattern of cortical atrophy in the early stages at the level of the fusiform gyrus and anterior regions of the right temporal lobe, which plays a key role in ensuring this function. This manifestation, in many patients, can also present in a multimodal manner, making it difficult to recognise familiar people not only by face but also by voice [67,68,69]. Spatial disorientation and episodic memory disturbances are other clinical features that distinguish rtvFTD. Topographic disorientation is defined as a condition in which the patient has difficulty recognising structures, buildings, and monuments that are typically used as references for orientation; this condition is particularly associated with lesions of the bilateral or right occipito-temporal regions. Another element that can be associated is anterograde disorientation, which is a selective disturbance in the acquisition of new information, typically related to degeneration of the right parahippocampal gyrus, located in close proximity to the region responsible for the regulation of topographic orientation; consequently, topographic and anterograde disorientation often coexist in the same subject [60,65,70,71]. The functions regulating language, which are usually spared at onset, are impaired as the pathology progresses, with the appearance of difficulty in lexical retrieval and frequent anomie. Moreover, it should be noted that the amnestic-prosopagnosic presentation of rtvFTD at onset may be easily confused with AD. However, all patients with rtvFTD would appear to have better performance on neuropsychological tests for the assessment of mnestic functions than those with AD [61,72]. In conclusion, rtvFTD, although it shares aspects with svPPA, bvFTD, and AD, also shows unique clinical features that are strictly dependent on the anatomical distribution of atrophy, which affects the right temporal lobe mainly in its anterior regions.

## 4. Genetics and Pathomechanism

FTD is a highly heritable group of neurodegenerative disorders with a vast genetical heterogeneity. Overall, about 40% of FTD patients have a positive family history of dementia, psychiatric diseases, or motor symptoms with 10% of them showing an autosomal dominant inheritance pattern [73]. However, the extent of heritability varies across the large spectrum of clinical phenotypes, with bvFTD (48%) and FTD-ALS (10–40%) showing stronger family history than PPA (12%) and svPPA being the least likely to have a genetic etiology [74]. To date, autosomal dominant mutations in *MAPT*, *GRN*, and *C9orf72* account for the majority of genetically determined FTD (about 30%), with large hexanucleotide (GGGGCC) repeat expansion in the first intron of *C9orf72* being the most common cause worldwide [73]. Recent evidence has identified TANK-binding kinase 1 (*TBK1*) as probably the fourth most common genetic cause of FTD [75]. Autosomal dominant mutations in many other genes cumulatively account for <5% of all FTD, and several genetic risk factors have been identified (see Table 1) [73]. In some isolated cases, an autosomal recessive inheritance pattern may be observed. Different case series have demonstrated geographical variability among those different genes [76,77,78]. Given this genetic heterogeneity, it is not surprising that the neuropathology underlying clinical FTD is also heterogeneous: TAR-DNA-binding protein-43 (TDP-43) proteinopathies, especially types A or B, are the commonest causes of genetic FTD, followed by tauopathies, fused-in-sarcoma (FUS) pathologies, and other rarer proteinopathies [79]. The neuropathology associated with specific mutations is shown in Table 1 and described in the dedicated paragraph [79]. As for other neurodegenerative diseases, FTD pathomechanism is hallmarked by protein misfolding leading to the formation of toxic extracellular or intracellular inclusions in neuronal and glial cells [80]. This neurodegenerative process is thought to originate in distinct regions and to spread within neural networks from cell to cell in a prion-like manner [81]. Several genetic hits (loss of function and toxic gain of function) contribute to alter proteostasis, boosting or sustaining vicious cycles that ultimately dysregulate pivotal cellular components (lysosome, mitochondria, and endoplasmic reticulum) or processes (autophagolysosomal trafficking, RNA homeostasis, endoplasmic reticulum-mitochondrial signalling, and axonal transport) and cause protein accumulation, sensitising cells to insult and finally leading to cell death [82,83,84,85,86]. Following the description of the molecular genetics of the three most common genetic causes, we will trace an overview of the key pathological mechanisms involved in FTD pathogenesis.

### 4.1. MAPT

In 1990, mutations in *MAPT* gene were associated with an autosomal inherited form of FTD with parkinsonism; this was the first evidence of a genetic cause for familial FTD [87]. *MAPT*, located in chromosome 17q21.31, encodes for microtubule-associated protein tau, a microtubule-associated protein co-assembling with tubulin into microtubules, particularly enriched in axons. The protein is involved in microtubule stabilisation and assembly (cytoskeletal dynamics) and axonal transport and regulates neuronal activity, neurogenesis, iron transport, and synaptic long-term depression [88]. It also localises in dendrites, where the function is unclear, and in the nucleus, where it might play a role in genomic DNA maintenance [88]. The *MAPT* gene is made up of 16 exons producing, by alternative mRNA splicing, six different tau isoforms that can contain three or four carboxy-terminal repeat domains (3R or 4R) associated with distinct tauopathies [88,89]. The term ‘tauopathies’ describes a range of clinically distinct neurodegenerative diseases pathologically hallmarked by hyperphosphorylated and insoluble aggregates of tau within neurons and glial cells [88]. To date, more than 80 mutations in the *MAPT* gene, either exonic or intronic, have been discovered and are associated with different tauopathies. Missense mutations cluster in the region encoding the microtubule-binding domain, thus reducing tau affinity for micro-tubules and increasing their tendency to form filaments or aggregate into neurofibrillary tangles [88]. Most splicing mutations are within intron 10, where the critical splicing generating the 3R and 4R isoforms occurs. The increase in exon 10+ mRNA raises the proportion of 4R tau isoform 4, consistent with the neuropathology initially described in families with FTD with parkinsonism [88]. The alteration of the 3R and 4R isoforms ratio, which intrinsically have different affinity to micro-tubules, triggers tau aggregation through mechanisms not fully understood [88]. Large gene rearrangements, such as deletions and duplications, have also been reported [89,90]. *MAPT* mutations cause neurotoxicity through three main mechanisms: (1) tau loss of function, which has usually been ascribed to aggregation and/or hyperphosphorylation, leading to microtubule disassembly and axonal transport deficit [86]; (2) tau neurotoxic gain of function, through formation of tau aggregates, leading to spatial interference with axonal transport, mitochondrial dysfunction, lysosome functioning, and impairment of autophagic pathways; in addition, tau aggregates, monomeric, hyperphosphorylated or mutant tau have been proposed as having direct toxic actions [83,86]; and (3) tau mislocalisation into post-synaptic spines, driven by tau hyperphosphorylation, mutations, and overexpression, leading to synaptic dysfunction [86,91]. Interestingly, apart from hyperphosporilation, other tau isoform post-translational modifications have been studied as drivers for the production of both oligomeric and soluble forms of tau [92]; for example, concerning proteolysis, studies have shown that proteolytic fragments of tau can be neurotoxic in a fragment-dependent manner as a result of intracellular aggregation and/or transcellular propagation [93]. The clinical presentation of patients harbouring *MAPT* pathogenic variants is heterogeneous even within the same family; bvFTD, nfvPPA, PSP, and CBS are all described, whereas symptoms of MND are rare [73,74,94,95].

### 4.2. GRN

The observation of many autosomal dominant FTD cases, not harbouring mutations in the *MAPT* gene but still genetically linked to the same chromosomal region (chr17q21), led to the discovery, in 2006, of the *GRN* gene as a novel causative disease gene [96]. *GRN* encodes for the glycoprotein progranulin, expressed in the central nervous system (CNS) by neurons and microglia, which after proteases-mediated cleavage into lysosomes produce smaller proteins named granulins. Progranulin and granulins act as a growth factor regulating cell cycle and motility via mitogen-activated protein kinase (MAPK) and Ak strain transforming (Akt) signalling pathways and are implicated in several biological processes such as angiogenesis, wound healing, inflammation, lysosomal function, brain development, and synapse functioning [97]. To date, more than 70 different mutations throughout the gene have been described, including frameshift, splicing, and nonsense mutations (resulting in a premature stop codon), as well as deletions. All of them cause disease via haploinsufficiency and lead to FTD with TDP-43 pathology [98]. Complete loss of progranulin causes a lysosomal storage group of neuropediatric disorders termed neuronal ceroid lipofuscinoses, characterised by seizures, dementia, visual loss, and cerebral atrophy. Thus, lysosomal dysfunction is crucial in GRN-FTD pathology, as observed in cellular and animal models and in patients with heterozygous *GRN* mutations [97]. Granulins are generated by lysosomal proteases and can themselves regulate the activity of several lysosomal enzymes maintaining lysosomal function [97]. Progranulin deficiency, also via impairment of ganglioside catabolism, alters lysosomal homeostasis, including lysosome trafficking, leading to neurodegeneration [97,99]. Notably, other rarer FTD-associated genes, including *VCP1*, *CHMP2B*, sequestosome 1 (*SQSTM1*), *TBK1*, and optineurin (*OPTN*), are involved in the autophagy–lysosome pathway [73,97]. Moreover, given its mitogenic and neurotrophic action and its role in regulating inflammatory response, loss of progranulin has been associated with excessive complement-dependent synaptic pruning, microglia hyperactivation after neural injury, and altered cytokine expression in serum and cerebrospinal fluid (CSF) [97,100]. The lysosome dysfunction itself, through gangliosidosis, may trigger the activation and recruitment of microglia, generating a mechanistic link between lysosomal metabolism and neuroinflammation [100]. From the clinical point of view, *GRN* mutations show incomplete penetrance and are associated with a vast phenotypic heterogeneity and diverse onset age. The most common clinical diagnoses are bvFTD and nfvPPA, the latter being distinct from the aphasic syndrome occasionally observed in patients with FTD-tau or the lvPPA in AD spectrum. CBS is also common, whereas symptoms of MND are rare. Extrapyramidal signs, in the form of CBS or dystonia, are present in 40% of cases, usually emerging after the onset of the core phenotype characterised by behavioural or language symptoms [95,96].

### 4.3. C9orf72

Ground-breaking progress in the genetics of FTD was made by the discovery of a genetic linkage between familial cases of FTD-MND and chromosome 9q21-q22. Exanucleotide GGGGCC repeat expansion in the first intron of the gene located in this locus, *C9orf72*, was recognised as the most common genetic cause of both diseases in Europe and North America [73,74,101]. This discovery demonstrated that FTD and ALS are intimately linked and share a common genetic and pathomechanistic background. *C9orf72* colocalises with several proteins implicated in autophagy and endosomal transport, playing pivotal roles in endosomal trafficking [102]. The vast majority of healthy subjects carries between 2 and 20 repeats, while larger expansions, from 100 to 1000, are commonly observed in patients with FTD-MND [73,74]. However, the current data do not support a single defined pathological repeat-length threshold, given the presence of somatic instability of the mutation and somatic mosaicism within the CNS [99]. Despite being in a noncoding region of the gene, the expanded hexanucleotide repeats are bidirectionally transcribed into repetitive RNAs via a noncanonical translation mechanism called RAN (repeat associated non-ATG), generating five different dipeptide repeat proteins (DPRs) [73,74,101]. Postmortem examination shows that *C9orf72* expansion carriers present brain TDP-43 positive inclusion, consistent with TDP-43 type B and A neuropathology, as well as sense and antisense RNA foci and DPRs inclusions [73,74,79]. The intronic hexanucleotide repeat expansions generate three different nonexclusive mechanisms that contribute to FTD-MND pathology with different extents, explaining the phenotypic heterogeneity observed in patients: (1) loss of function of C9orf72 protein impairs synaptic vescicle recycling, causes lysosomal accumulation, alters glial secretomes (contributing to neuroinflammation), and affects autophagy-mediated RNA homeostasis [81,82,83]; (2) toxic gain of function of RNA foci: GGGGCC repeat RNAs form secondary structures (i.e., hairpins, stable G-quadruplexes, DNA–RNA heteroduplexes, and RNA duplexes), which affect promoter activity, genetic instability, RNA splicing, localisation, and transport system (through the sequestration of RNA-binding proteins) [81,82,83,85]; and (3) toxic gain of function of DRPs: arginine-rich DRPs have been shown to alter nucleocytoplasmic transport impacting neuronal survival as for RNA foci [81,82,83,85]. Interestingly, a strong interplay between three FTD-ALS-associated genes, *C9orf72*, *TBK1*, and *TARDBP*, was recently found: *TBK1* is phosphorylated in response to C9orf72 poly(gly-ala) aggregation and sequestered into inclusions, resulting in decreased TBK1 activity and contributing to neurodegeneration. Reducing *TBK1* activity in mice exacerbated poly(gly-ala)-induced phenotypes, including increased TDP43 pathology and the accumulation of defective endosomes in poly(gly-ala)-positive neurons. Thus, a disruption of the endosomal-lysosomal pathway in FTD-ALS leads to increased susceptibility to protein aggregation, driving TDP43 proteinopathy and neurodegeneration [103]. The most common phenotype associated with *C9orf72* repeat expansion is bvFTD, followed by MND and FTD-MND [94,95]. Age of onset and disease progression are highly variable; for example, *C9orf72* mutations can present with a slowly progressive bvFTD phenotype [104]. In familial FTD-MND cases, *C9orf72* repeat expansion accounts for the disease in more than 50% of families. *C9orf72* repeat expansion carriers manifesting with FTD show high frequency of psychosis, delusions, and hallucinations when compared with other FTD gene carriers [73,74,94,95].

**Table 1 ijms-24-11732-t001:** FTD-causative genes and genetic risk factors.

**Causative Genes**							
**Gene**	**Chromosome**	**Mutations**	**Frequency**	**Protein Function**	**Neuropathology**	**Phenotypes**	**References**
*C9orf72*	9	Intronic hexanucleotide repeat expansions	10%	Nucleocytoplasmic transport, autophagy, intercellular trafficking	TDP-43 (types B, A), RNA foci, DRPs inclusions	FTD, ALS, FTD-ALS	[73,74,81,82,83,85]
*GRN*	17	Frameshit, splicing, nonsense, deletions	10%	Angiogenesis, wound healing, inflammation, lysosomal function, brain development, synapse functioning	TDP-43 type A	FTD, PPA, CBS	[73,74,96,97,99,100]
*MAPT*	17	Missense, splicing, deletions, duplications	10%	Microtubule stabilisation, assembly, neuronal activity, neurogenesis, iron transport, DNA maintenance	TAU	FTD, FTD with parkinsonism, PSP, CBS, AD	[73,74,86,87,88,89,105]
*TBK1*	12	Missense	5%	Pattern recognition receptors signalling pathway upon viral infection, autophagy	TDP-43 types A, B	ALS, FTD, FTD-MND	[103,106,107]
*TARDBP*	1	Missense	1%	Encodes for TDP-43, RNA processing and metabolism, stress granule formation	TDP-43	ALS, FTD with or without MND, FTD-MND plus hypokinetic or hyperkinetic movement disorders	[108,109]
*FUS*	16	Missense	1%	DNA and RNA metabolism, including DNA repair, transcription regulation, RNA splicing and export to the cytoplasm	FET	ALS	[110]
*CHMP2B*	3	Splicing	<1%	Encodes for a component of ESCRT-III (endosomal sorting complex required for transport III), degradation of surface receptor proteins, formation of endocytic multivesicular bodies	UPS	FTD, ALS, FTD-MND	[111]
*VCP-1*	9	Missense	<1%	Organelle biogenesis, ubiquitin-dependent protein degradation, autophagy	TDP-43 type D	IBMPFD (Paget bone disease, inclusion body myositis and FTD), ALS, FTD-MND (MSP)	[112,113]
*SQSTM1*	5	Missense	<1%	NFkB signaling, apoptosis, transcription regulation, ubiquitin-mediated autophagy	TDP-43	Paget bone disease, ALS, FTD, distal myopathy (MSP)	[114]
*CHCHD10*	22	Missense	<1%	OXPHOS regulation, maintenance of mitochondrial cristae morphology	Not classified, no TDP-43 accumulation	FTD-MND, mitochondrial myopathy	[115]
*OPTN*	10	Missense, deletions	<1%	Vesicular trafficking, endocytic trafficking, NFkB signaling	TDP-43 type A	ALS, FTD-MND	[107]
*UBQLN2*	X	Missense	<1%	Regulation of proteasome-mediated ubiquinated proteins degradation	U	ALS with FTD	[116]
*TUBA4A*	2	Missense	<1%	Microtubule network assembly	TDP-43 type A	ALS, FTD	[117]
*CCNF*	16	Missense	<1%	Proteasomal degradation	TDP-43	FTD-ALS	[118]
*TIA1*	2	Missense	<1%	Splicing regulation, translation repression	TDP-43	ALS, FTD-ALS	[119]
*CYLD*	16	Missense	<1%	Deubiquitination, negative regulator of NFkB	TDP-43	FTD-ALS	[120]
*ABCA7*	19	Frameshit	<1%	Lipids transporter, phagocytosis		FTD	[75]
*CTSF*	11	Missense	<1%	Lysosomal protease		FTD	[75]
**Risk Factors**							
**Gene**	**Chromosome**	**SNPs**	**Protein Function**		**Details**		**References**
*TMEM106B*	7	rs102004, rs6966915, rs1990622	Transmembrane protein, late endosomes and lysosome functioning		Increased TMEM106B expression level (rs1990622 protect GRN and C9orf72 mutations carriers from developing FTD)		[121,122]
*RAB8/CTSC* locus	11	rs302652	Protein trafficking to lysosomal-related organelles, maturation of phagosomes/serine proteinases activation in immune and inflammatory response		50% reduction in RAB8 mRNA level in blood		[123]
HLA locus	6	rs1980493	Immune system regulation		Changes in the methylation levels related to HLADRA in the frontal cortex		[123]
*GFRA2*	8	rs36196656	Neuronal differentiation, proliferation and survival		Decreased GFRA2 expression level		[124]

## 5. Neuropathology

Despite their considerable clinical heterogeneity, FTD syndromes usually present with degeneration mainly affecting the frontal and/or temporal lobes associated with microscopic changes, such as neuronal loss, synaptic alterations, microvacuolations, and gliosis [125]. Notably, these pathological changes are not specific to FTD. Indeed, it is possible to highlight alterations characteristic of FTD in a wide range of disorders, such as AD [126]. Based on the molecular aspects of the inclusions in neurons and glial cells, FTD neuropathology can be divided into three main groups: TDP-43 (about 50% of cases), tau (about 40%), and FET, including FUS, EWSR1 (Ewing sarcoma breakpoint region 1/EWS RNA binding protein 1), and TAF15 (TATA-box-binding protein-associated factor 15), accounting for 10% of cases [127,128]. In particular, in the different FTD variants:bvFTD: atrophy mainly affects the anterior cingulate cortex, anterior insula, striatum, amygdala, hypothalamus, and thalamus [129]. These interconnected areas form a salience network that allows us to focus attention on the internal and external stimuli of interest [130]. Approximately 15–20% of bvFTD results from mutations in the genes described above (*MAPT*, *GRN*, and *C9orf72*), resulting in slightly different patterns of alterations [8]. In particular, *MAPT* mutations result in a ventral degeneration pattern (initially affecting the amygdala, hippocampus, entorhinal cortex, and temporal pole) [131], whereas *GRN* mutations are associated with a lateral degeneration pattern [132], with frequent extension to areas not typical of bvFTD, particularly in the posterior regions, possibly for an overlap with AD pathology [133]. Regarding the *C9orf72* hexanucleotide repeat expansion, the most common genetic cause of FTD, a mild but diffuse damage could be present [134], or a prevalence in the medial thalamus or, rarely, a cerebellar involvement [135]. In sporadic bvFTD, the most common neuropathological findings are Pick’s disease, FTLD-TDP (type B in particular), and corticobasal degeneration. As for FTD genetic forms, *MAPT* mutations cause a specific tauopathy, whereas *GRN* mutations are generally associated with FTLD-TDP type A. *C9orf72* expansion also causes FTLD-TDP, but the subtype is less regular (however, type B is the most frequent) [128].svPPA: it is characterised by marked atrophy of the left anterior temporal lobe, progressively extending to the contralateral temporal lobe and the orbitofrontal and posterior brain areas [72]. In the late stages, atrophy is also evident in the cingulate cortex, thalamus, and hippocampal region [136]. It generally shows C-type TDP-43 inclusions, characterised by long dystrophic neurites [136,137], but also combinations of different protein aggregates, including TDP type A and B, tau, β-amyloid, and α-synuclein pathology, have been described [138,139,140].nfvPPA: the anterior areas of the language circuit are the most vulnerable, and in particular, the dominant inferior frontal lobe is almost always involved. Moreover, other areas of the dominant hemisphere, in particular the anterior opercular and perisylvian ones, the anterior insula, and the superior temporal gyrus, are often affected [36]. Tau-positive inclusions are most commonly found [137], but TDP-43 and, in cases where agrammatism is not emphasised, even AD pathology have been highlighted [1].rtvFTD: atrophy usually starts in the right temporal lobe and then spreads to either the frontal or left temporal areas [141]. The most common findings are FTLD-TDP type C, tau-MAPT, and TDP type A and B. On the other hand, svFTD, i.e., its left counterpart, is associated with the TDP type C pathology [137,141,142]. Furthermore, associations with FUS [79] and TDP-E have very rarely been described [141,143]. Interestingly, this pathology is often associated with MND [144].

## 6. Diagnosis

### 6.1. Diagnostic Criteria of FTD

Early and accurate diagnosis of FTD turns out to be crucial from a prognostic, therapeutic, and patient management point of view, as do the implications arising from possible heritability of the disorder. The diagnostic criteria for bvFTD accepted and used to date are clinical and neuroradiological criteria, developed and reviewed by the International Behavioural Variant FTD Criteria Consortium (FTDC) in 2011, which provide for the identification of core features of the disorder and allow for a diagnosis of possible, probable, or definite bvFTD [26]. The diagnosis of possible bvFTD is based solely on the clinical syndrome, in order to identify patients in the early stages of the disorder. It requires the presence of at least three out of six cardinal symptoms: disinhibition, apathy/inertia, loss of sympathy/empathy, perseverative/compulsive behaviours, hyperorality, and a dysexecutive neuropsychological profile. Instead, the diagnosis of probable bvFTD is based on clinical manifestations and, in addition, a demonstrable functional decline and aspects to neuroimaging that show cortical changes typical of bvFTD. In addition, an exclusion criterion for diagnosis is the presence of biomarkers that are strongly suggestive of AD or other neurodegenerative disorders. On the other hand, the definite diagnosis is exclusively provided in the case of patients who present with clinical syndrome bvFTD and have pathogenetic mutations or histopathological evidence of FTLD [26]. The diagnostic criteria for PPA, on the other hand, go back to the work of Mesulam et al., who predicted that there is an isolated speech disorder prominent during the early phase of the disease, with an insidious onset and gradual progression [2]. Autonomy in activities of daily living is generally spared, except for those activities that involve the use of language. Finally, aphasia should be the prominent disorder at the onset of the disease, during its initial stages, and at the time of clinical evaluation. Exclusion criteria include prominent impairment of episodic and nonverbal memory and visuospatial functions at onset, the presence of specific causes of aphasia (e.g., stroke), which must be excluded by neuroimaging, and the presence of behavioural alterations as a pivotal symptom at onset. Once these criteria are met, the diagnostic process for PPA involves classification into one of three variants of the disorder, according to the work by Gorno-Tempini et al. in 2011. It is therefore necessary to consider the presence or absence of salient aspects of language, considering specific domains represented by speech production features, repetition, single-word and syntax comprehension, confrontation naming, semantic knowledge, and reading/spelling. The classification of PPA into one of the variants may occur at one of three levels: clinical, imaging-supported, or definite pathologic diagnosis [145].

### 6.2. Neuroimaging

Lately, a great effort has been made to find early biomarkers that can lead to a pre-symptomatic diagnosis of FTD. Multimodal imaging has been proven to be a valid assessment tool [146]. Brain magnetic resonance imaging (MRI) and 18F-fluoro-2-deoxyglucose positron emission tomography ([18F]FDG-PET) scans together have a sensitivity of 96% and a specificity of 73% [147]. MRI is often the first exam to be performed and may show atrophy in the frontal and/or temporal lobes, which can support the diagnosis of FTD [26,145]. In patients without early MRI changes, [18F]FDG-PET seems a sensitive marker for pre-symptomatic diagnosis of bvFTD [147]. Moreover, hypometabolism seems to be more diffuse at baseline with a higher annual change than volume loss [148]. In particular:bvFTD: bvFTD patients show bilateral medial prefrontal, right orbitofrontal, anterior insular cortex, and anterior cingulate cortex atrophy [1,149]. A smaller grey matter volume has been found in superior, middle, and inferior frontal gyrus, orbito-frontal, insular, temporal, and parahippocampal gyrus and hippocampus, compared to controls [150]. Furthermore, in bvFTD, there are significant volume reductions in striatum, bilateral globus pallidus, and left putamen [1]. Through resting-state functional MRI (fMRI), it is possible to show that patients with bvFTD have a preferential disruption of the intrahemispheric connectivity, in particular in the frontoinsular, temporal, and basal ganglia networks bilaterally [151]. The salience network (involving frontal-insula-anterior cingulate gyrus) also displays decreased functional connectivity [149]. Surprisingly, enhanced connectivity has been observed among the basal ganglia and relatively unaffected regions, although it is yet to be explained if this is a cause or a consequence of the disease [151]. Hypometabolism in FDG-PET mainly involves caudate nuclei, superior medial frontal cortex of both sides, right middle frontal gyrus and right inferior frontal cortex, left anterior cingulate cortex, and right inferior temporal gyrus [152]. Usually, it spreads from the frontal regions into parietal and temporal cortices [148].svPPA: in svPPA patients, the anterior temporal lobe atrophy is bilateral, usually asymmetrical, and typically left sided. Over time, it may involve the posterior temporal lobes and the inferior frontal lobes [149]. Volume reduction in the amygdala has been observed too [1]. White-matter atrophy has been shown in the temporal portions of the inferior longitudinal fasciculus, inferior fronto-occipital fasciculus, and uncinate fasciculus bilaterally [137]. Hypometabolism has been demonstrated in the temporal lobe, with an asymmetric anteroposterior gradient (posterior more than anterior) in lateral temporal regions, mainly observed in the left hemisphere [148]. Metabolic decline has also been observed in bilateral anterior medial temporal, orbitofrontal, medial prefrontal, medial and inferolateral parietal cortices, and subcortical structures [148].nfvPPA: the atrophy in nfvPPA patients usually involves the anterior perisylvian cortex of the dominant hemisphere, in particular the left frontal operculum and Broca’s areas 44, 45, and 47 [149]. Over time, the atrophy expands into the left precentral, inferior, and middle frontal gyri, anterior insula, inferior parietal cortex, and subcortical structures [148]. White-matter atrophy has been observed, in particular involving the left superior longitudinal fasciculus and the body of the corpus callosum, as well as bilateral anterior corona radiata [137]. Hypometabolism has been demonstrated in the same areas by FDG-PET [1].rtvFTD: it has recently been introduced into FTD’s clinical syndromes. Consensus criteria for the diagnosis still need to be defined [55], and there are still few studies that describe the pattern of atrophy in this variant. MRI shows grey-matter volume loss of the right ventral frontal area and the left temporal lobe, similarly to svPPA. In particular, an MRI shows bilateral asymmetrical (right more than left) grey-matter atrophy in the anterior temporal lobes and in the right ventral frontal area. Right-sided grey-matter atrophy has been observed in the temporal poles, the superior, medial, and inferior temporal gyri, medial temporal lobe, insula, fusiform gyrus, angular gyrus, supramarginal gyrus, inferior frontal gyrus, gyrus rectus, and orbitofrontal cortex [61]. As the disease progresses, thinning in the orbitofrontal cortex and anterior cingulate has been reported [72].

Absence of amyloid binding, because of its high negative predictive value, rules out AD in favour of other causes of dementia, such as bvFTD [153]. A very promising role could be played by the new tau protein ligands, which are currently used almost exclusively in the research framework. Interestingly, [18F]AV-1451 uptake has been demonstrated in svPPA in the temporal lobe and ventromedial frontal regions, which are usually the atrophic areas in this disease. Furthermore, nfvPPA patients show uptake in frontal and basal ganglia regions and in the white matter of the frontal lobes. More studies are needed in this direction [154].

#### Neuroimaging in Genetic FTD

About a third of FTD cases may be ascribed to different genetic autosomal-dominant mutations [155]. Patients with genetic-transmitted FTD show the first symptoms even in early adulthood [156], but perhaps more importantly, there is evidence of atrophy and hypometabolism at least 5–10 years before the expected onset of symptoms in asymptomatic genetic mutation carriers [157]. In the *C9orf72* mutation carriers, the early structural imaging changes may be detected 20 years before the onset of symptoms [157]. These patients show earlier atypical FTD atrophy patterns compared to other FTD gene mutation carriers, even though the patterns look typical when compared with healthy noncarriers. The regions that show lower grey-matter volume are the cerebellum, thalamus, insula, and cortical frontal and temporal regions, with stable reduction over time [1]. White-matter integrity changes are often observed in the anterior thalamic radiation [74]. Resting-state fMRI describes involvement of the salience network and a medial pulvinar thalamus-seeded network [74]. *GRN* mutation carriers show presymptomatic asymmetrical atrophy, involving the frontal, parietal, and insular cortexes, as well as the striatum [74] and cortical thinning in the right temporal lobe [158]. Insula is the first affected area, followed by the temporal and parietal lobes, while the earliest subcortical involvement is in the striatum [157]. White-matter changes emerge in an anterior–posterior internal capsule. Moreover, it has been shown that white-matter hyperintensities are characteristic of *GRN* mutation [74]. The analysis of resting-state fMRI reveals involvement of the fronto-parietal network [74]. *MAPT* mutation carriers show a more symmetric pattern [1], involving the temporal lobe and medial temporal structures, such as the hippocampus and amygdala [157]. White-matter degeneration affects the uncinate fasciculus [158] and parahippocampal cingulum [74]. Resting-state fMRI shows involvement of the default mode network [74]. FDG-PET does not have a specific pattern in genetic FTD, as it shows hypometabolism in the grey-matter areas of atrophy [74]. Moreover, currently, tau protein radioligands have also not been shown to be helpful [74].

### 6.3. Biomarkers

A biomarker can be defined as a reliable index of a disease condition. In FTD, the usefulness of early identification is linked both to the poor correlation between clinical phenotype and underlying neuropathology and to the possibility of designing drug trials for particular disease subgroups (e.g., patients affected by a specific mutation) [159]. In the context of neurodegenerative diseases, it is common to study the CSF, since it communicates directly with the neuronal interstitium, and therefore, the concentration of molecules in the CNS environment is higher than in peripheral blood [160]. Moreover, the examination of peripheral blood samples is associated with further difficulties: the presence of blood proteolytic enzymes [161], antibodies [162], high concentrations of various proteins that can interfere with measurements [163], and finally, the possibility that values may be elevated because of alterations in systems other than the CNS for proteins also expressed in other body regions [159]. Despite these technical problems, the introduction of highly sensitive and specific immuno- and mass-spectrometry-based assays has made it possible to re-evaluate the use of blood samples to investigate neurodegenerative processes and obtain useful disease biomarkers [159,164]. In particular, the current research focuses on (see Figure 1 for main FTD biomarkers and their pathomechanisms):

Neurofilaments (NfLs): NfLs are a particular type of intermediate filaments and are fundamental components of the cytoskeleton in both the CNS and the periphery [165,166]. They are crucial in ensuring the stability of axons (especially the larger myelinised ones) [167], mitochondria, and the cytoskeletal content of microtubules [168,169]; at the synaptic level, they guarantee the structure and function of dendritic spines and glutamatergic and dopaminergic neurotransmission [170]. NfLs increase in many neurological diseases, reasonably because of damage and degeneration of axons [171], resulting in their release into the CSF and, subsequently, into the blood, in which they are usually present at a ratio of 1:40 to the CSF [172]. They seem to be able to identify FTD patients, especially those affected by bvFTD [173], as well as being a possible index of disease severity, since they correlate with survival [174]. Interestingly, values in the CSF correlate well with those in the blood [175], and values in presymptomatic subjects are lower than in symptomatic ones, allowing a possible follow-up [176]. Specifically, NfLs are above normal limits in all categories of FTD, i.e., bvFTD, nfvPPA, and svPPA [177], with the exception of lvPPA [178,179]. Furthermore, the increase in NfL values is more pronounced in FTD patients than in other neurodegenerative diseases, including AD [180], LBD, mixed dementia (vascular and AD-related), Parkinson’s disease dementia (PDD), and other types, with useful implications for differential diagnosis [181,182]. Moreover, given the close correlation between CSF and blood values of NfLs, measurements on blood samples have also been shown to distinguish FTD patients from healthy controls [183,184,185]. Another important aspect to emphasise is the ability of high values of NfLs to distinguish between FTD and primary psychiatric disorders, which is often clinically demanding [153,186]. Both NfL values in the CSF [187] and blood values [183] show utility from a prognostic point of view, as values at baseline correlate with the progression of cognitive deficits, assessed by the mini-mental-state examination (MMSE) and clinical dementia rating (CDR), as well as with survival [179,188]. Regarding the genetic forms of FTD, the highest values of NfLs in the CSF were shown in association with the *GRN* mutation [189], whereas the highest blood values were documented in patients with the *C9orf72* expansion [190]. However, some potential concerns associated with the use of this biomarker must be considered: first, it is not sufficient on its own to make the diagnosis of FTD [191]; moreover, although it is generally believed that an increase in this biomarker reflects axonal damage, it could also be attributable to increased transport by exosomes or through active secretion [192] or reflect an alteration at the synaptic rather than the axonal site [193]. In addition, their drainage may occur along the intramural perivascular spaces and/or by lymphatic and glymphatic systems, so the mechanisms of transport from the CNS have not yet been fully elucidated [194]. Finally, the reason why levels are higher in FTD than in other types of dementia is not yet fully understood, as it could be due to a higher severity of FTD in terms of neurodegeneration [165] or subclinical motor neuron degeneration linked to a concomitant ALS, especially in a TDP-43 pathology [195]. Even considering these caveats, their diagnostic and prognostic value is clear, and further studies may delve into the still unresolved issues, also to design future clinical trials.TDP-43: TDP-43 is a highly conserved nuclear RNA/DNA-binding protein crucial for RNA processing regulation [196]. TDP-43-positive cytoplasmic inclusions are shown in about 50% of FTD patients, mostly in bvFTD [197] and in svPPA, sometimes in FTD-MND, and rarely in nfvPPA [159]. In particular, in bvFTD, the spread of the phosphorylated-TDP-43 (p-TDP-43) pathology encompasses four stages: in the first one, there are p-TDP-43 inclusions in the basal and anterior portions of the prefrontal neocortex and amygdala; in the second stage, p-TDP-43 spreads in the anteromedial area, superior and middle temporal gyri, and subsequently, striatum, and medial and lateral portions of the thalamus. In the third stage, the pathological burden is present in the motor cortex, neocortical areas, and spinal cord anterior horn. In the final stage, the inclusions spread to the occipital neocortex [195]. Notably, the spreading pathway is very similar to that of ALS, providing interesting clues about a possible pathological overlap [195]. Unfortunately, most of the CSF TDP-43 amount is due to the passage through the blood–brain barrier (BBB); thus, CSF levels do not reflect the precise neuropathological condition in the CNS [198]. Of note, both plasma and CSF levels of p-TDP-43 are higher in patients carrying the C9orf72 mutation, in comparison with other genetic variants [199]. Very interestingly, recent work by Scialò et al. highlighted the possibility of using RT-QuIC to identify the presence of TDP-43 on CSF, exploiting both the excellent level of technology achieved and the prion-like behaviour of the protein aggregates, providing considerable insight into the early detection of this finding in patients with ALS and FTD [200]. Furthermore, a very recent study showed the possibility of using a multimer detection system to assess the plasma oligomeric form of TDP-43, highlighting a significant increase in patients with svPPA compared to healthy controls and other neurodegenerative diseases, suggesting its usefulness as a plasma biomarker [201]. However, it is important to point out that TDP-43 forms various types of assemblies (e.g., monomers, dimers, oligomers, and aggregates), whose significance, in terms of function, phase separation, and aggregation, is not yet fully understood [202]. Further studies are therefore needed to clarify the role of this complex protein in the pathophysiology of FTD.Progranulin: it is a ubiquitous growth factor, which is important for tissue development, proliferation, and repair [159]; in particular, progranulin has been implicated in various brain mechanisms [203], including neurite outgrowth [204], stress response [205], TDP-43 aggregation [206], and synaptic function [207], although the evidence is not yet conclusive. It is reduced in the CSF of patients with bvFTD and svPPA (i.e., those with predominantly TDP-43 pathology) compared to those with nfvPPA (i.e., those with predominantly tau pathology) [159,208]. Importantly, patients with GRN mutations manifest reduced progranulin concentrations in both blood and CSF; thus, this index could be used to identify carriers of this mutation in the appropriate clinical context [209,210]. Moreover, the reduction in its levels is associated with complement activation in brain tissue, demonstrated by increases in complement fractions C1qa and C3B in the CSF during the disease course [211].β-amyloid and tau: according to the recent ATN classification, the presence of β-amyloid and tau (in CSF and/or in neuroimaging) is the neuropathological hallmark of AD [212]. β-amyloid, in particular, generally might help to rule out other dementias in the differential diagnosis, although overlaps in the neuropathological frame are sometimes observed, leading to inconsistencies between the clinical diagnosis and the neuropathological classification [213,214,215,216]. Thus, FTDs have lower levels of the secreted form of the amyloid precursor protein [217], and the combined use of altered NfL values and normal values of β-amyloid 42 allows patients with FTD to be differentiated with good sensitivity and specificity from AD ones or healthy controls [180]. The FTD subtype showing the lowest amount of typical AD biomarkers is bvFTD [218], and regardless of variant, all FTD patients have a lower ratio of phosphorylated tau (p-Tau) to total tau (t-Tau) [179]. The only exception is the so-called lvPPA, which, consistently, has neuropathological findings compatible with AD in the majority of cases and is now more commonly related to AD than FTD [57]. Concerning plasma markers, some studies have reported higher t-tau levels in patients with bvFTD and PPA compared to healthy controls [219], while a recent meta-analysis has shown that AD patients have higher p-tau values than those with FTD, underlining their potential role in the differential diagnosis [220].Glial fibrillary acidic protein (GFAP): astrogliosis, i.e., the inflammatory reaction against damage that characterises glial cells in various neuropathological contexts, including neurodegenerative diseases, can be assessed by measuring related markers, such as GFAP [221]. Interest in this marker is growing, as shown by recent studies that have documented a correlation between its plasma levels and β-amyloid pathology, and not tau pathology, in AD patients [222]. A recent literature review has shown altered levels of this marker, in particular in the plasma of subjects characterised by the GRN mutation and with higher levels in symptomatic patients than in presymptomatic ones, highlighting a potential prognostic role of this protein [220].Protein triggering receptor expressed on myeloid cells 2 (TREM2): TREM2 is an innate immunity receptor that characterises microglial cells, and its expression increases during phagocytosis, response to neuronal damage, and chemotaxis [223]. Therefore, it could be used as a marker of microglial activity in patients affected by FTD [159], and its soluble fraction (sTREM2) is measurable in both CSF and blood [199]. In particular, its CSF levels are elevated in GRN patients, whereas no significant differences have been documented between patients with FTD variants and healthy controls [224].Dipeptide repeats: the increased expression of polyglutamine (poly(GP)) linked to the expansion of C9orf72 is a characteristic feature of most familial forms of FTD [81]. This expansion results in the production of aberrant proteins (i.e., abnormal DPRs), which can be found in the CSF of patients with a specificity of 100% [178]. Notably, their levels are particularly high in symptomatic mutation carriers; thus, it might be a potential marker of disease activity [159].

#### Promising Biomarkers

Considering the limitations of the biomarkers examined so far, research is currently underway to identify more reliable ones. For instance, a recent study examined the soluble plasma form of the suppressor of tumourigenicity 2 (ST2), a highly brain-expressed interleukin 33 receptor, showing, compared to healthy controls, the highest values in FTD patients, followed by AD and Parkinson’s disease [225]. Furthermore, considering that tau protein can be cleaved into different fragments that are actively secreted by the cells in the CSF, Foiani et al. examined their potential as biomarkers in FTD diagnosis, without, however, demonstrating superior diagnostic accuracy compared to other available biomarkers [226]. However, as the authors point out, the different disease groups examined had different patterns in the concentrations of the different fragments, and it is possible that not all existing tau fragments were evaluated. Therefore, despite the negative result so far, it is possible that further studies may demonstrate the superiority of some tau fragments over others in the diagnosis of FTD. Finally, very interestingly, to investigate the molecular mechanisms responsible for the different pathological accumulations of FTLD, recent work has used a mass-spectrometry-based proteomic and systems-level analysis of the middle frontal gyrus cortices, highlighting cyclin-dependent kinase 5 and polypyrimidine-tract-binding protein 1 as key players in the disease process [227]. Of note, the pathological alterations were associated with changes in astrocyte and endothelial cell protein abundance levels, highlighting that the changes typical of the disease are not limited to neurons and that glial cells may also provide valuable information on the pathophysiology of FTD.

### 6.4. Neurophysiology

In recent years, novel neurophysiological techniques, such as transcranial magnetic stimulation (TMS), have been investigated in neurodegenerative diseases to better characterise some aspects of cortical circuitry, plasticity, reactivity, and connectivity [228,229]. Undoubtedly, the most investigated pathology has been AD, in which several correlations with neuropsychological and neuropathological biomarkers have been made, showing that AD is characterised by an impairment of cortical plasticity, hyperexcitability, and altered connectivity [230], mirroring the neuropathological studies in animal models [231]. The use of these techniques might be able not just to allow a better understanding of the neurophysiological changes occurring in neurodegenerative diseases but also to serve as potential outcome measures for clinical trials. Converging evidence shows that FTD is associated with changes in several neurotransmitter systems, particularly in GABAergic and glutammatergic circuits, while cholinergic transmission seems to be unaffected [232,233]. Conversely to AD, no significant changes in excitability have been reported [234,235,236]. Intracortical circuitry in FTD patients has been extensively investigated, reporting the main differences with AD; indeed, many studies have shown alterations in protocols investigating the inhibitory and facilitatory intracortical circuitry, namely short-interval intracortical inhibition (SICI), long-interval intracortical inhibition (LICI), and intracortical facilitation (ICF) [157,237,238,239], possibly reflecting the GABAergic and glutammatergic deficiencies that are associated with FTD. Specifically, a reduction in SICI and LICI highlighted the impairment of GABAergic transmission for both the main receptor activities (GABA-A and GABA-B), resulting in a disinhibition of intracortical circuitry, while the impairment of ICF is consistent with an impairment of facilitation caused by glutammatergic N-methyl-D-aspartate (NMDA) receptor involvement. Notably, central cholinergic activity, as measured by the short-afferent inhibition (SAI) protocol, has turned out to be a sensitive marker in the AD spectrum, even for early stages [240], and has been shown to be activated during memory tasks [241]. Importantly, the SAI protocol did not show abnormalities in FTD patients, congruently with the fact that, in FTD pathology, there is a relative sparing of the cholinergic system [235,238]. Interestingly, TMS measures have been able to show good power in differentiating patients affected by different neurodegenerative diseases [242], even at prodromal stages [243].

## 7. FTD Animal Models

Of note, much insight into FTD pathophysiology has been gained in recent decades through studies in animal models, which have also made it possible to assess the therapeutic potential of new drugs in animals carrying specific mutations [244]. For example, *Caenorhabditis elegans* carries the human homologues of the *MAPT*, *GRN*, *VCP*, and *TDP-43* genes, making it an excellent model for understanding the molecular aspects of the disease and for developing new therapeutic targets [244,245]. Furthermore, as in many other diseases, the low cost and rapid life cycle of *Drosophila melanogaster*, which carries the human homologues of VCP, TDP-43, and tau, has been exploited [244]. For example, transgenic forms of Drosophila characterised by *MAPT* mutations that are pathological for humans have made it possible to identify mechanisms that suppress tau toxicity [246]. Similarly, zebra fishes (*Danio rerio*) have numerous and highly conserved genes associated with human neurological diseases, making it possible to assess the molecular mechanisms linked to neurodegeneration [247]. However, transgenic mice are the animals currently most used as models, as they have neural networks similar to humans and also allow studies to be carried out at the molecular and gene level [244]. Indeed, transgenic mice carrying specific mutations in genes associated with FTD have been developed, which also present behavioural modifications similar to the corresponding symptoms in humans (e.g., apathy, disinhibition, socio-emotional alterations, and memory and eating-behaviour disorders) [248]. To date, the most studied models include transgenic mice for tau, GRN, TDP-43, and C9orf72 [249,250,251,252]. Interestingly, some animal models also make it possible to study the role of neuroinflammation in FTD, which is believed to play a key role in a large number of neurodegenerative diseases [253].

## 8. Treatments

Despite efforts and numerous discoveries in recent years, there is still no treatment for FTD patients [13]. Several pharmacological trials are currently underway, mainly in the field of familial FTD with autosomal dominant mutations but also some for sporadic forms [55]. For example, in patients with tau-related FTD, a therapeutic option is passive immunisation with anti-tau monoclonal antibodies, in order to improve the clearance of this protein and to counteract its prion-like spread [55,254]. Moreover, experimental studies are underway on monoclonal antibodies directed against sortilin, a protein involved in the degradation of progranulin in patients with mutations causing progranulin deficiency [255]. Other innovative gene-editing therapies deal with antisense oligonucleotides to counteract excessive RNA expression in patients with the *C9orf72* mutation [256]. Moreover, non-invasive brain stimulation techniques (NIBS) are increasingly widespread and promising not only from a diagnostic point of view (see Section 6.4) but also from a therapeutic perspective, since they allow the modulation of specific rhythms and neuronal populations not only locally but also at the network level by exploiting anatomical connections and functional synchronisation [257,258]. In recent years, increasing evidence strongly supports the key role of neuroinflammation in the pathophysiology of FTD and other neurodegenerative diseases. Indeed, neuroinflammation may alter synaptic transmission contributing to the progression of neurodegeneration, as largely documented in animal models and in studies on patients [259]. Recently, a pilot study showed beneficial effects in a sample of FTD patients treated with palmitoylethanolamide, an endogenous lipid mediator known to exert a modulatory effect on neuroinflammation processes that are present during the neurodegenerative process [260]. However, at the moment, physical, occupational, and speech therapies, together with proper caregiver education, improvement of the environment, and support provided by the health services and dedicated associations, are the cornerstone in patient management [13,261,262]. Finally, symptomatic therapies can be used to try to control symptoms that impair the patient’s quality of life, causing a complex management [55,263]. For example, considering the serotonin deficiency in FTD patients, serotonin re-uptake inhibitors (SSRIs) are a good option for depressive symptoms [264], while neuroleptics (especially atypical ones, given their better risk profile compared to typical ones) improve cognitive and behavioural symptoms, reducing delusions, psychomotor agitation, and consequently, the caregiver burden [265,266,267].

## 9. Conclusions

Although still considered a rare disease, given the low incidence of cases per year, FTD is increasingly gaining awareness in clinicians and researchers thanks to the recent developments and knowledge in some pathological mechanisms. Improvement in the clinical diagnosis of AD has also allowed a better definition of those patients who do not have an AD pathology, increasing the clinician’s confidence in formulating different diagnoses. In this review, we have highlighted the main clinical and diagnostic clues for an FTD (and its variants) diagnosis, including also some perspective on novel biomarkers that could help not just for diagnosis but also for investigation of pathophysiological mechanisms and for possible outcome measures in future clinical trials. We hope that it will provide physicians and researchers with an overview of the current evidence on this complex topic from a clinical, diagnostic, and therapeutic point of view, offering insights for crucial future research in this field. Figure 2 summarises the key clinical, neuropathological, and neuroimaging features of the main variants of FTD.

## Figures and Tables

**Figure 1 ijms-24-11732-f001:**
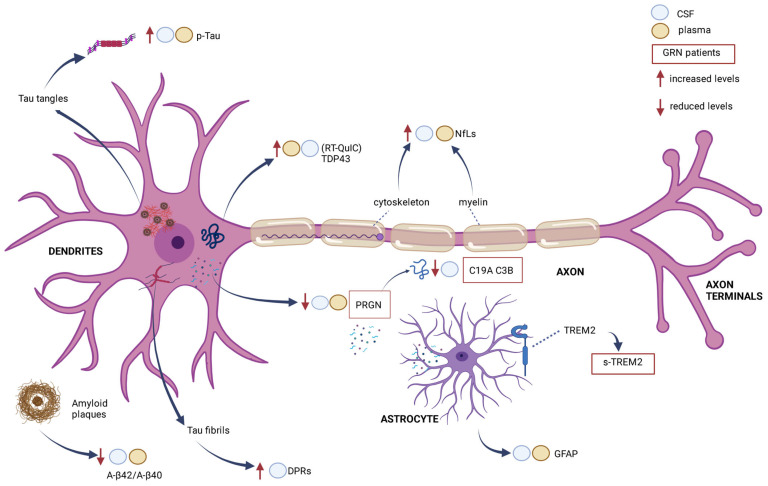
Main FTD biomarkers and their pathomechanisms. CSF: cerebrospinal fluid; DPRs: dipeptide repeat proteins; GFAP: glial fibrillary acidic protein; GRN: progranulin; NfLs: neurofilaments; RT-QuIC: real-time quaking-induced conversion; s-TREM2: soluble fraction of protein triggering receptor expressed on myeloid cells 2; TREM2: protein triggering receptor expressed on myeloid cells 2.

**Figure 2 ijms-24-11732-f002:**
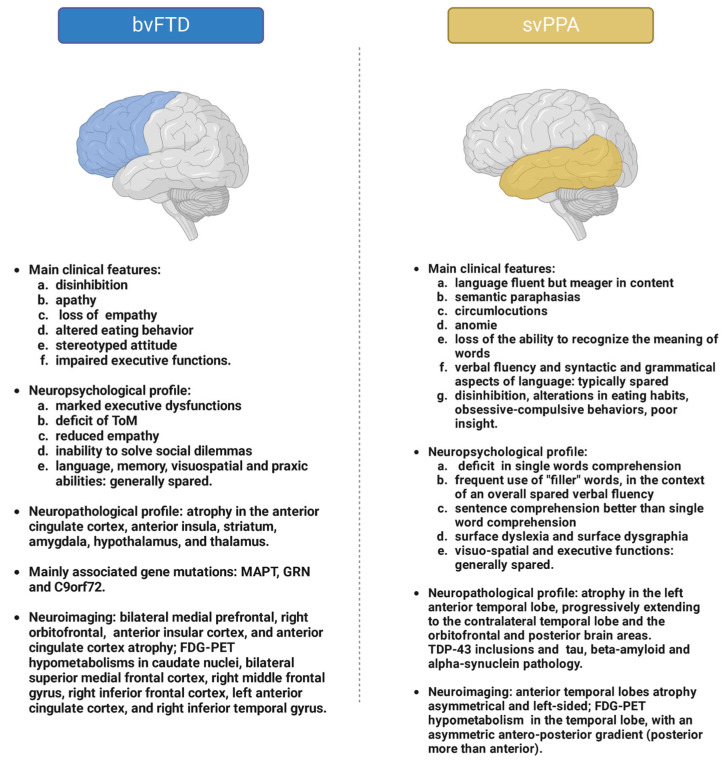
Main clinical, neuropsychological, neuropathological, and neuroradiological features of FTD variants.

## Data Availability

No new data were created or analysed in this study. Data sharing is not applicable to this article. All the literature used for this review is listed in the bibliography.

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
