# Peer review of "Frontotemporal Dementia, Where Do We Stand? A Narrative Review"

_ijms, 2023, doi:10.3390/ijms241411732_

Round 1
Reviewer 1 Report
Dear Authors,
I was invited to review your review entitled “Frontotemporal Dementia, where do we stand? A narrative review” (Manuscript ijms-2505369).
This manuscript provides in a detailed manner a state of art on frontotemporal dementia (FTD), a neurodegenerative disorder characterized by multiple and distinct neuropathological changes that feature different forms of this disorder.
The manuscript is written and organized very well and is very detailed in each paragraph.
For this purpose I would suggest to draw a figure that shows all forms of FTD, highlighting the similarities and differences between them (i.e. age, gender, anatomical locations, genetic mutations, biomarkers, clinical manifestations, etc.). In this manner I think that the comprehension of the text becomes more simple.
I have some curiosities:
-Today, are there animal models to study the different pathological forms of FTD?
-You say that mutations of tau protein (expecially hyperphosphorylation) and its aggregation are responsable for the development of some forms of FTD.
What about other post-translational modifications, for example truncation with the conseguent generation of toxic fragments?
The paragraph 4.3 C9orf72 presents many errors of agreement between the subject and the predicate and in line 558 correct C9ORF72 with C9orf72.
Correct the following errors:
-line 36: add (ALS);
-line 65: add the number for the reference (Mesulam et al.) and eventually change the numeration of all references;
-line 194: substitute “by” with “of” these individuals;
-line 533, 534, 536, 543, 546, 549, 550: errors of agreement between the subject and the predicate;
-line 581: correct microscipic with microscopic; synpatic with synaptic; microvaculations with microvasculations;
-line 581: correct characteristics with characteristic;
-line 596: correct pole with lobe;
-line 650: correct muations with mutations;
-line 651: add the number of the reference (Mesulam et al.);
-line 661: add the number of the reference Gorno-Tempini et al., 2011;
-line 691: correct hypometabolisms with hypometabolism;
-line 749: correct presymtomatic with presymptomatic;
-line 753: correct matte with matter;
-line 781: correct 781 with filaments with filament;
-line 840: correct trhough with through;
-line 1000: correct NDMA with NMDA;
-line 1013: correct Tubulin Alpha 4° with Tubulin Alpha 4°.
Regarding the reference list, please:
-delete ePub, PMID and PMIC;
-add doi when absent;
add authors at 34, 40, 41, 42, 43, 44, 46, 47, 48, 52, 53, 54, 56 references;
-adjust all bibliography according the IJMS guidelines.
The title is clear and concise.
The english language is good.
For these reasons I think that the manuscript needs to minor revisions for the publication of the manuscript.
The english language is good.
Author Response
Dear Reviewer,
thank you very much for your work and valuable advice, that have enabled us to greatly improve the quality of our paper.
We have revised the manuscript according to your suggestions, in particular:
- we have added a figure summarizing the clinical, neuropsychological, neuropathological, and neuroimaging features of the different variants of FTD (please see Figure 2)
- we have added a short paragraph in the text about animal models of FTD (please see paragraph 7. FTD animal models):
“7. FTD animal models
Of note, much insight into FTD pathophysiology has been gained in recent decades through studies in animal models, which have made it possible also to assess the therapeutic potential of new drugs in animals carrying specific mutations [244]. For example, Caenorhabditis elegans carries the human homologues of the MAPT, GRN, VCP, and TDP-43 genes, making it an excellent model for understanding the molecular aspects of the disease and for developing new therapeutic targets
[244,245]. Furthermore, as in many other diseases, the low cost and rapid life cycle of Drosophila melanogaster, which carries the human homologues of VCP, TDP-43, and tau, has been exploited [244]. For example, transgenic forms of Drosophila characterised by MAPT mutations that are pathological for humans have made it possible to identify mechanisms that suppress tau toxicity [246]. Similarly, Zebra fishes (Danio rerio) have numerous and highly conserved genes associated
with human neurological diseases, making it possible to assess the molecular mechanisms linked to neurodegeneration [247]. However, transgenic mice are the animals currently most used as models, as they have neural networks similar to humans and also allow studies to be carried out at the molecular and gene level [244]. Indeed, transgenic mice carrying specific mutations in genes associated with FTD have been developed, which also present behavioural modifications similar to
the corresponding symptoms in humans (e.g. apathy, disinhibition, socio-emotional alterations, memory and eating behaviour disorders) [248]. To date, the most studied models include transgenic mice for tau, GRN, TDP-43, and C9orf72 [249-252]. Interestingly, some animal models make it possible also to study the role of neuroinflammation in FTD, which is believed to play a key role in a large number of neurodegenerative diseases [253]”.
- we have added a sentence and a new paragraph about promising biomarkers in the context of future study and treatment prospects, in which there are some explanatory sentences about tau post-translational modifications, please see:
“Interestingly, apart from hyperphosporilation, other tau isoforms post-translational modifications have been studied as driver for the production of both oligomeric and soluble forms of tau [92]; for example, concerning proteolysis, studies have shown that proteolytic fragments of tau can be nerutoxic in a fragment-dependent manner as a result of intracellular aggregation and/or transcellular propagation [93]”.
“Furthermore, considering that tau protein can be cleaved into different fragments that are actively secreted by the cells in the CSF, Foiani et al. examined their potential as biomarkers in FTD diagnosis, without, however, demonstrating superior diagnostic accuracy compared to other available biomarkers [226]. However, as the Authors point out, the different disease groups examined had different patterns in the concentrations of the different fragments and it is possible that not all existing tau fragments were evaluated. Therefore, despite the negative result so far, it is
possible that further studies may demonstrate the superiority of some tau fragments over others in the diagnosis of FTD”.
- we have corrected all the errors in the text according to your instructions
- we revised and adjusted the reference list as you indicated and according to the IJMS guidelines.
Reviewer 2 Report
Dear Authors,
On the review paper on FTD would be of great interests for the field.
Authors summarized the important biomarkers and their contributions.
However, authors left out several latest reports as listed below.
In addition, authors should include few graphics on the biomarkers and their mechanisms.
1. Clusters of co-abundant proteins in the brain cortex associated with fronto-temporal lobar degeneration. Bridel C, van Gils JHM, Miedema SSM, Hoozemans JJM, Pijnenburg YAL, Smit AB, Rozemuller AJM, Abeln S, Teunissen CE. Alzheimers Res Ther. 2023 Mar 23;15(1):59. doi: 10.1186/s13195-023-01200-1.
2. Plasma Soluble ST2 Levels Are Higher in Neurodegenerative Disorders and Associated with Poorer Cognition. Tan YJ, Siow I, Saffari SE, Ting SKS, Li Z, Kandiah N, Tan LCS, Tan EK, Ng ASL. J Alzheimers Dis. 2023;92(2):573-580. doi: 10.3233/JAD-221072.
3. Increased oligomeric TDP-43 in the plasma of Korean frontotemporal dementia patients with semantic dementia. Jamerlan AM, Shim KH, Youn YC, Teunissen C, An SSA, Scheltens P, Kim S. Alzheimers Dement. 2023 May 18. doi: 10.1002/alz.13127.
4. Shapeshifter TDP-43: Molecular mechanism of structural polymorphism, aggregation, phase separation and their modulators. Doke AA, Jha SK. Biophys Chem. 2023 Apr;295:106972. doi: 10.1016/j.bpc.2023.106972.
Authors need to revise the verb tenses.
Author Response
Dear Reviewer,
thank you very much for your work and valuable advice, that have enabled us to greatly improve the quality of our paper. We have revised the manuscript according to your suggestions, in particular:
- we have added a figure regarding the main FTD biomarkers and their pathomechanisms, please see Figure 1
- we have added to the reference list the most recent work you pointed out to and added explanatory sentences about it in the text, including a new paragraph entitled “Promising biomarkers”, please see:
“Furthermore, a very recent study showed the possibility of using a multimer detection system to assess the plasma oligomeric form of TDP-43, highlighting a significant increase in patients with svPPA compared to healthy controls and other neurodegenerative diseases, suggesting its usefulness as a plasma biomarker [201]. However, it is important to point out that TDP-43 forms various types of assemblies (e.g. monomers, dimers, oligomers, aggregates), whose significance, in terms of function, phase separation, and aggregation, is not yet fully understood [202]. Further studies are therefore needed to clarify the role of this complex protein in the pathophysiology of FTD”.
“6.3.1 Promising biomarkers
Considering the limitations of the biomarkers examined so far, research is currently
underway to identify more reliable ones. For instance, a recent study examined the soluble plasma form of Suppressor of tumourigenicity 2 (ST2), a highly brain-expressed interleukin 33 receptor, showing, compared to healthy controls, the highest values in FTD patients, followed by AD and Parkinson's disease [225]. Furthermore, considering that tau protein can be cleaved into different fragments that are actively secreted by the cells in the CSF, Foiani et al. examined their potential as biomarkers in FTD diagnosis, without, however, demonstrating superior diagnostic accuracy compared to other available biomarkers [226]. However, as the Authors point out, the different disease groups examined had different patterns in the concentrations of the different fragments and it is possible that not all existing tau fragments were evaluated. Therefore, despite the negative result so far, it is
possible that further studies may demonstrate the superiority of some tau fragments over others in the diagnosis of FTD. Finally, very interestingly, to investigate the molecular mechanisms responsible for the different pathological accumulations of FTLD, recent work used a mass spectrometry-based proteomic and systems-level analysis of the middle frontal gyrus cortices, highlighting cyclin-dependent kinase 5 and polypyrimidine tract-binding protein 1 as key players in the disease process [227]. Of note, the pathological alterations were associated with changes in astrocyte and endothelial cell protein abundance levels, highlighting that the changes typical of the disease are not limited to neurons and that glial cells may also provide valuable information on the pathophysiology of FTD”.
- we finally revised the language, with the support of an expert in the field.